



# Improving weather and climate predictions by training of supermodels

Francine Schevenhoven[1,2], Frank Selten[3], Alberto Carrassi[4,1,2], and Noel Keenlyside[1,2]

[1]Geophysical Institute, University of Bergen, Bergen, Norway
[2]Bjerknes Centre for Climate Research, Bergen, Norway
[3]Royal Netherlands Meteorological Institute, De Bilt, The Netherlands
[4]Nansen Environmental and Remote Sensing Center, Bergen, Norway

**Correspondence:** F.J. Schevenhoven (francine.schevenhoven@uib.no)

**Abstract.**

Recent studies demonstrate that weather and climate predictions potentially improve by dynamically combining different models into a so called "supermodel". Here we focus on the weighted supermodel - the supermodel's time derivative is a weighted superposition of the time-derivatives of the imperfect models, referred to as weighted supermodeling. A crucial step

is to train the weights of the supermodel on the basis of historical observations. Here we apply two different training methods to a supermodel of up to four different versions of the global atmosphere-ocean-land model SPEEDO. The standard version is regarded as truth. The first training method is based on an idea called Cross Pollination in Time (CPT), where models exchange states during the training. The second method is a synchronization based learning rule, originally developed for parameter estimation. We demonstrate that both training methods yield climate simulations and weather predictions of superior quality as

compared to the individual model versions. Supermodel predictions also outperform predictions based on the commonly used Multi-Model Ensemble (MME) mean. Furthermore we find evidence that negative weights can improve predictions in cases where model errors do not cancel (for instance all models are warm with respect to the truth). In principle the proposed training schemes are applicable to state-of-the-art models and historical observations. A prime advantage of the proposed training schemes is that in the present context relatively short training periods suffice to find good solutions. Additional work needs to

be done to assess the limitations due to incomplete and noisy data, to combine models that are structurally different (different resolution and state representation for instance) and to evaluate cases for which the truth falls outside of the model class.

## 1 Introduction

### 1.1 Premises and the multi-model-ensemble

Although weather and climate models continue to improve, they will inevitably remain imperfect (Bauer et al., 2015). Nature

is so complex that it is impossible to model all relevant physical processes solely based on the fundamental laws of physics (think for instance about the micro-physical properties of clouds that determine the cloud radiational properties). Progress in predictive power crucially depends on further improving our knowledge and the numerical representation of the physical

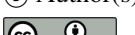



processes the model is intended to describe. Nevertheless, with the best possible models in hands, more accurate predictions can be obtained by making good use of all of them thus exploiting multi-model information. In order to reduce the impact of model errors on predictions, it is common practice to combine the predictions of a collection of different models in a statistical fashion. This is referred to as the Multi-Model Ensemble (MME) approach: the MME mean prediction is often more

skillful as model errors tend to average out (Weigel et al., 2008), whereas the spread between the model predictions is naturally interpreted as a measure of the uncertainty about the mean (IPCC, 2013). Although MME tends to improve predictions of climate statistics (i.e. mean and variance), a major drawback is that it is not designed to produce an improved trajectory that can be seen as a specific climate forecast, given that averaging uncorrelated climate trajectories from different models leads to variance reduction and smoothing.

The foundation of modern weather and climate prediction rests on the assumption that when an estimate of the climate state is at disposal at a particular instance in time, its time evolution can be calculated by a proper application of a numerical discretization of the fundamental laws of physics, supplemented by empirical relationships describing unresolved scales and a complete specification of the external forcing and boundary conditions. Integration in time subsequently yields a predicted climate trajectory into the future, and formally frames the climate prediction endeavor as a mixed initial and boundary condi-

tions problem (see e.g. Collins and Allen, 2002; Hawkins and Sutton, 2009). Initial conditions, but also boundary conditions and external forcing are usually estimated by combining data with models via data assimilation techniques (see e.g. Carrassi et al., 2018, for a review). Errors in the time-derivative (i.e. the model error) propagate into errors in the predicted trajectory but model error also affects the model statistics, so that the model and observed mean and variance differ, giving rise to model biases.

An illustrative example of this propagation of model errors is presented in Rodwell and Jung (2008) in relation to a change in the model's prescribed aerosol concentrations in the region of the Sahara. Already, within the first few hours of prediction, the different aerosol concentration leads to changing the stability and convection in the region. This in turn changes the upper air divergence and promotes the generation of large-scale Rossby waves that travel horizontally eastward and northward into the Northern Hemisphere during the subsequent week and finally impact the surface air temperatures in Siberia. This example

evidences that a specific model error can impact model prediction skills on far regions and diverse variables. Furthermore it suggests that, in order to mitigate or in the best case to prevent model error from growing and affecting the whole model phase space, it is better to intervene at each model computational time step rather than a posteriori by combining outputs after a prediction is completed as in the MME approach.

## 1.2 Supermodeling

Reducing model errors early in the prediction is precisely what supermodeling attempts to achieve (van den Berge et al., 2011). In a supermodel, different models exchange information during the simulation at every time step and form a consensus on a single best prediction. An advantage over the standard MME approach is that the supermodel produces a trajectory with improved long term statistics. Improved trajectories are extremely valuable for calculations of the impact of climate on society.





For instance crop yields, spread of diseases, river discharges all depend on the specific sequences of weather events, not just on statistics (Challinor and Wheeler, 2008; Sterl et al., 2009; van der Wiel et al., 2019).

The supermodeling approach was originally developed using low-order dynamical systems (van den Berge et al., 2011; Mirchev et al., 2012) and subsequently applied to a global atmosphere model (Schevenhoven and Selten, 2017; Wiegerinck and Selten, 2017) and to a coupled atmosphere-ocean-land model (Selten et al., 2017). A partial implementation of the super-modeling concept using real world observations was presented in Shen et al. (2016). In the original supermodeling concept, model equations are connected by nudging terms such that each model in the ensemble is nudged to the state of every other model at every time step. For appropriate connections the ensemble of models eventually synchronizes on a common solution that depends on the strength of the connections. For instance, if all models are nudged to a particular model that is not nudged to any other model, the ensemble will follow that particular solution. By training connections on observed data, an optimal solution is found that is produced by the connected ensemble of models. This type of supermodel is referred to as *connected* supermodeling. Wiegerinck and Selten (2017) showed that in the limit of strong connections the connected supermodel solution converges to the solution of a weighted superposition of the individual model equations, referred to as a *weighted* supermodel.

A crucial step in supermodeling is the training of the connection coefficients (for connected supermodels) or weights (for weighted supermodels) based on data, the observations. The first training schemes of supermodels were based on the minimization of a cost function dependent on long simulations with the supermodel (van den Berge et al., 2011; Mirchev et al., 2012; Shen et al., 2016). Given that iterations, and thus many evaluations of the cost function, were necessary in the minimization procedure, this approach turned out to be computationally very expensive. Schevenhoven and Selten (2017) developed a computationally very efficient training scheme based on Cross Pollination in Time (CPT), a concept originally introduced by Smith (2000) in the context of ensemble weather forecasting. In CPT, the models in a multi-model ensemble exchange states during the simulation, generating mixed trajectories that exponentially increase in number in the course of time. As a consequence, a larger area of phase space is explored thus increasing the chance that the observed trajectory is shadowed within the span of all of the mixed model trajectories. Given the above, CPT training is then based on the selection of the trajectory that remains closest to an observed trajectory. Another alternative efficient approach or training was introduced in Selten et al. (2017) to learn the connections coefficients in a supermodel. Their method, hereafter referred to as *synch rule* is based on synchronization and it is inspired by an idea originally proposed in Duane et al. (2007) for general parameter learning.

Before supermodeling becomes suitable for the class of large dimensional state-of-the-art weather and climate models, we need to have training schemes that are computationally suitable for that context. In this paper we develop, apply and compare CPT and the synch rule to train a weighted supermodel based on the intermediate complex global coupled atmosphere-ocean-land model SPEEDO (Severijns and Hazeleger, 2010). Short-term supermodel prediction skill as well as long-term climate statistics show that both training methods result in supermodels that outperform the individual models. Furthermore, original experiments with negative weights, as opposed to the standard case of weights larger or equal to zero, suggest that even when the individual model biases do not compensate for each other an improved supermodel solution can be achieved.

In Sect. 2, the two types of supermodels, *connected* and *weighted*, are introduced in detail. Section 3 describes the global coupled atmosphere-ocean-land model SPEEDO and the construction of a SPEEDO supermodel. The two training strategies





are described in Sect. 4 with specific details when applied to the SPEEDO model in Sect. 5. The results of the training are shown in Sect. 6. The final section discusses the results and lists further steps to be taken towards training a supermodel based on state-of-the-art weather and climate models using real world observations.

## 2 Weighted and connected supermodeling

To make the supermodeling approach more explicit, we formally write the model equations of a weather or climate model $i$ as

$$\dot{\mathbf{x}}_i = \mathbf{f}_i(\mathbf{x}_i, \mathbf{p}_i) \tag{1}$$

where $\mathbf{x}_i$ is a high-dimensional state vector, $\mathbf{f}$ a non-linear evolution function depending on the state $\mathbf{x}_i$ and on a number of adjustable parameters $\mathbf{p}_i$. In practice, weather and climate models generally differ in the representation of the climate state, i.e. the phase where $\mathbf{x}_i$ is defined, the evolution function and parameter values. In this stage of developing the supermodeling

approach and training schemes we simplify the context and focus on a situation where the models share the same evolution function, $\mathbf{f}$, and the same phase space, so that $\mathbf{x}_i \in \mathbb{R}^n$ for all $i$. However, the models differ in the parameters, $\mathbf{p}_i \neq \mathbf{p}_j$ if $i \neq j$. The approach can be generalized using data assimilation approaches (Du and Smith, 2017). We will furthermore denote the *truth* as given by the model $\mathbf{f}$ with a specific set of parameters. An ensemble of imperfect models can be dynamically combined in a *weighted* or *connected* supermodel.

### 2.1 Weighted supermodeling

A weighted supermodel based on two imperfect models is given by

$$\dot{\mathbf{x}}_1 = \mathbf{f}(\mathbf{x}_s, \mathbf{p}_1) \tag{2a}$$
$$\dot{\mathbf{x}}_2 = \mathbf{f}(\mathbf{x}_s, \mathbf{p}_2) \tag{2b}$$
$$\dot{\mathbf{x}}_s = \mathbf{W}_1 \dot{\mathbf{x}}_1 + \mathbf{W}_2 \dot{\mathbf{x}}_2, \tag{2c}$$

where $\mathbf{x}_s \in \mathbb{R}^n$ represents the supermodel state vector and diagonal matrices $\mathbf{W}_1 = \mathrm{diag}(\mathbf{w}_1)$ with $\mathbf{w}_1 \in \mathbb{R}^n$ denote the weights. In the weighted supermodel the states are imposed to be perfectly synchronized. Training a weighted supermodel implies training the weights $\mathbf{w}_i$.

### 2.2 Connected supermodeling

For completeness and for comparison of the weighted supermodels with the connected supermodel from Selten et al. (2017),

we introduce the equations for the connected supermodel. A connected supermodel based on two imperfect models is given by





$$\dot{\mathbf{x}}_1 = \mathbf{f}(\mathbf{x}_1, \mathbf{p}_1) - \mathbf{C}_{12}(\mathbf{x}_1 - \mathbf{x}_2) \tag{3a}$$

$$\dot{\mathbf{x}}_2 = \mathbf{f}(\mathbf{x}_2, \mathbf{p}_2) - \mathbf{C}_{21}(\mathbf{x}_2 - \mathbf{x}_1) \tag{3b}$$

$$\dot{\mathbf{x}}_s = \frac{1}{2}(\dot{\mathbf{x}}_1 + \dot{\mathbf{x}}_2), \tag{3c}$$

Note the nudging terms (the rightmost terms in Eq. 3a and Eq. 3b) that push the state of each model to the state of the other at every time step. The size of the nudging terms $\mathbf{C}_{12}$ and $\mathbf{C}_{21}$ reflects the strength of the coupling between the two models. They have the form of diagonal matrices $\in \mathbb{R}^{n \times n}$ and can thus be written as $\mathbf{C}_{12} = \mathrm{diag}(\mathbf{c}_{12})$ with $\mathbf{c}_{12} \in \mathbb{R}^n$. The diagonal form reflects the fact that each model state vector component is nudged towards the same component of the other model. The approach can be extended to be multivariate allowing for cross nudging, but this will require careful scaling of the variables.

For appropriate connections the models fall into a synchronized motion (Pecora and Carroll, 1990). Because in general the synchronization will not be perfect due to the different parameter values, the supermodel solution $\mathbf{x}_s$ is defined as the average of the different model states. Note that the states will be close for strong connections so that smoothing and loss of variance due to the averaging will be limited. The supermodel solution depends on the relative strengths of connection coefficients. Training a connected supermodel implies training the value of the connection coefficients.

A connected supermodel allows for more flexibility in case the ensemble is not perfectly synchronized (Wiegerinck et al., 2013). In regions of phase space of strong divergence for instance, one model can pull the ensemble along if it takes a very different trajectory. However, in Wiegerinck et al. (2013) it is noted that the size of the connection coefficients after training is typically quite large. The larger the coefficients, the stronger the models converge on a synchronized trajectory, which can be described by a weighted superposition of the models (Wiegerinck et al., 2013). Since for some training applications, perfect

synchronization is required as we shall see in Sect. 4, only weighted supermodels are considered in this paper. We do not limit ourselves to combining only two imperfect models into a supermodel, also combining four imperfect models will be discussed.

## 3   SPEEDO climate model

The SPEEDO global climate model consists of an atmospheric component (SPEEDY) that exchanges information with a land (LBM) and an ocean-sea-ice component (CLIO) using coupling routines (Fig. 1). The coupling routines perform re-gridding

operations between the computational grids of the different modules. A detailed description of SPEEDO can be found in Severijns and Hazeleger (2010); Selten et al. (2017).

The atmospheric model SPEEDY describes the evolution of the two horizontal wind components $U$ (east-west) and $V$ (north-south), temperature $T$ and specific humidity $q$ at eight levels in the vertical levels and the surface pressure $p_s$. Relatively simple calculations of heating and cooling rates due to radiation, convective transports, cloud amounts, precipitation and turbulent heat,

water and momentum exchange at the surface are performed at a computational grid of approximately 3.75 degree horizontal spacing (48x96 grid cells).





SPEEDY exchanges water and heat with the land model LBM that uses three soil layers and up to two snow layers to close the hydrological cycle over land and a heat budget equation that controls the land temperatures. The horizontal discretization is the same as for the atmosphere model. The land surface reflection coefficient for solar radiation is prescribed using a monthly climatology. Each land bucket has a maximum soil water capacity. The runoff is collected in river-basins and drained into the ocean at specific locations of the major river outflows.

SPEEDY exchanges heat, water and momentum with the ocean model CLIO (Goosse and Fichefet, 1999). CLIO describes the evolution of ocean currents, temperature and salinity on a computational grid of 3 degree horizontal resolution and 20 unevenly spaced layers in the vertical. A three-layer thermodynamic-dynamic sea-ice model describes the evolution of sea-ice in case ocean temperatures drop below freezing levels. Heat storage in the snow-ice system is accounted for and snow amounts and ice thickness evolve in response to surface and bottom heat fluxes. Sea ice is considered to behave as a viscous-plastic continuum as it moves under the action of winds and ocean currents.

Formally, the SPEEDO equations can be written as

$$\dot{\mathbf{a}} = \mathbf{f}^a(\mathbf{a}; \mathbf{p}^a) + \mathbf{g}^a(\mathbf{e}^h, \mathbf{e}^w, \mathbf{e}^m) \tag{4a}$$

$$\dot{\mathbf{o}} = \mathbf{f}^o(\mathbf{o}; \mathbf{p}^o) + \mathbf{g}^o(\mathcal{P}^o\mathbf{e}^h, \mathcal{P}^o\mathbf{e}^w, \mathcal{P}^o\mathbf{e}^m, \mathcal{P}^o\mathbf{r}) \tag{4b}$$

$$\dot{\mathbf{l}} = \mathbf{f}^l(\mathbf{l}; \mathbf{p}^l) + \mathbf{g}^l(\mathcal{P}^l\mathbf{e}^h, \mathcal{P}^l\mathbf{e}^w, \mathbf{r}), \tag{4c}$$

where $\mathbf{a}$ is the atmospheric state vector, $\mathbf{o}$ the ocean/sea-ice state vector, $\mathbf{l}$ the land state vector, $\mathbf{e}^h$ the heat exchange vector between atmosphere and surface, $\mathbf{e}^w$ the water exchange vector, $\mathbf{e}^m$ the momentum exchange vector and $\mathbf{r}$ the river outflow vector describing the flow of water from land to ocean. The exchange vectors depend on the state of the atmosphere and the surface but this dependency is not made explicit in Eq. 4 to simplify the notation. The projection operators $\mathcal{P}$ represent the re-gridding operations between the computational grids. These operations are conservative so that the globally integrated heat and water loss of the atmosphere at any time at the surface equals the integrated heat and water gain of the land and ocean. The non-linear functions $\mathbf{f}$ represent the cumulative contribution of the modelled physical processes to the change in the climate state vector and depend on the values of the parameter vectors $\mathbf{p}$. Some of these parameters go through a daily and/or seasonal cycle and/or have a spatial dependence like the reflectivity of the surface. The non-linear functions $\mathbf{g}$ describe how the exchange of heat, water and momentum between the subsystems affects the change of the climate state vector.



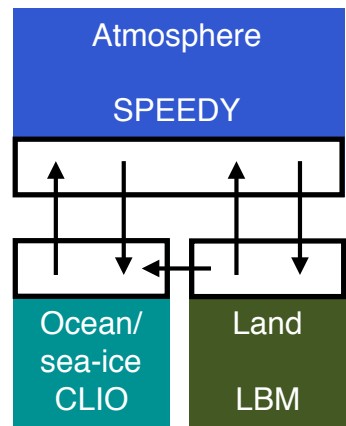

**Figure 1.** Schematic representation of the SPEEDO climate model. The atmosphere needs surface characteristics (temperature, roughness, reflectivity, soil moisture) in order to calculate the exchange of heat, water and momentum. Coupler software communicates this information between the components and interpolates between the computational grids.

## 3.1 SPEEDO supermodel

The training experiments of this study are evaluated in a noisy free observation framework, with perfect observations generated by sampling a reference model trajectory. This "perfect model" provides a set of time-ordered observations, called the "truth". We consider the SPEEDO climate model with standard parameter values as truth and create imperfect models by perturbing

parameter values in the atmospheric component. A supermodel is formed by combining the imperfect atmosphere models through a weighted superposition of the time derivatives of the imperfect models (Eq. 2) which are each coupled to the same ocean and land model (Fig. 2). All atmosphere models receive the same state information from the ocean and land model but each calculates their own water, heat and momentum exchange. On the other hand, the ocean and land model receive the multi-model weighted average of these atmospheric components; this follows the interactive ensemble approach (Kirtman and

Shukla, 2002). Following Eq. 2 , the SPEEDO weighted supermodel equations are given by

$$\dot{\mathbf{a}}_i = \mathbf{f}^a(\mathbf{a}_s; \mathbf{p}_i^a) + \mathbf{g}^a(\mathbf{e}_i^h, \mathbf{e}_i^w, \mathbf{e}_i^m) \tag{5a}$$

$$\dot{\mathbf{o}} = \mathbf{f}^o(\mathbf{o}; \mathbf{p}^o) + \mathbf{g}^o(\mathcal{P}^o\overline{\mathbf{e}^h}, \mathcal{P}^o\overline{\mathbf{e}^w}, \mathcal{P}^o\overline{\mathbf{e}^m}, \mathcal{P}^o\mathbf{r}) \tag{5b}$$

$$\dot{\mathbf{l}} = \mathbf{f}^l(\mathbf{l}; \mathbf{p}^l) + \mathbf{g}^l(\mathcal{P}^l\overline{\mathbf{e}^h}, \mathcal{P}^l\overline{\mathbf{e}^w}, \mathbf{r}) \tag{5c}$$

$$\dot{\mathbf{a}}_s = \sum_i \mathbf{W}_i \dot{\mathbf{a}}_i, \tag{5d}$$

where $\dot{\mathbf{a}}_s$ denotes the time derivative of the supermodel, $\mathbf{W}_i$ denote diagonal matrices with weights on the diagonal, $i$ refers to imperfect model $i$ and the overbar denotes a weighted average over the models.





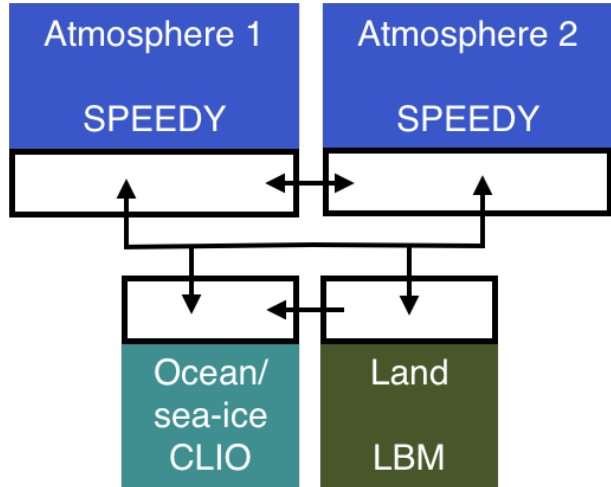

**Figure 2.** Schematic representation of the SPEEDO climate supermodel based on two imperfect atmosphere models. The two atmosphere models exchange water, heat and momentum with the perfect ocean and land model. The ocean and land model send their state information to both atmosphere models. The atmosphere models exchange state information in order to combine their time-derivatives.

## 4    Learning methods

Two different learning strategies are evaluated in this study in order to train the SPEEDO weighted supermodel: learning based on CPT as developed and applied to low-order dynamical systems in Schevenhoven and Selten (2017) and learning based on synchronization as applied to a connected SPEEDO supermodel in Selten et al. (2017).

### 5    4.1    Cross pollination in time

The Cross Pollination in Time (CPT) learning approach is based on an idea proposed by Smith (2001). CPT "crosses" trajectories of different models in order to create a larger solution space. The aim is to generate trajectories that follow the truth more closely. The training phase of CPT starts from an observed initial condition in state space. For simplicity assume the model is one dimensional. From the same initial state, the imperfect models compute one time step each ending in a different state. Next, all models compute another time step from each of these new states. Continuing this process leads to a rapid increase in the number of trajectories with time (Fig. 3a) that will ultimately cover a larger area of the state space. Among the full set of mixed trajectories, the one which is closest to the truth (i.e. to the data) is continued, the others are discarded resulting in a pruned ensemble, as is depicted in Fig. 3b.



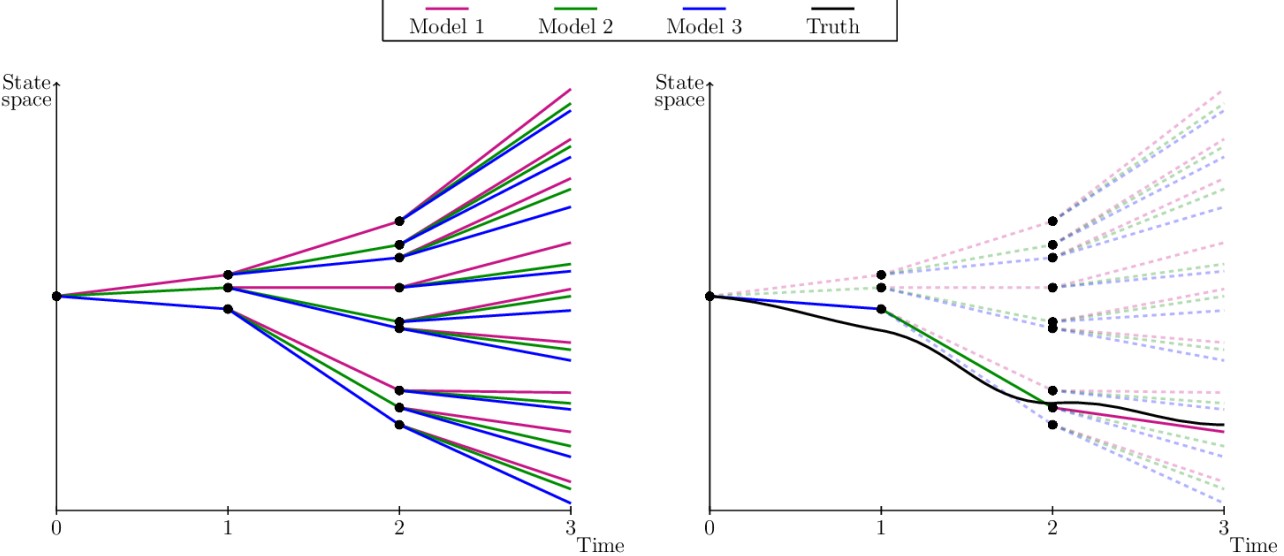

**Figure 3.** Adapted from Schevenhoven and Selten (2017). A one dimensional schematic of CPT for 3 models, a full ensemble (left) and a pruned ensemble (right). Note that the *truth* has been drawn here as a continuous line for illustrative purpose. In practice the *truth* is only known at discrete times (the observation times) and the distance with respect to model trajectories is computed at those times only.

In case of a multi dimensional model, such as SPEEDO, it is possible that at each time step different models are closest to the truth for different state variables and at different grid locations. In this case we continue per state variable with the model that is closest. This means that the initial state for the next time step can consist of a combination of models. As the values for the different state variables might not be in agreement with each other, this creates imbalances that can lead to numerical

5    instabilities. A (partial) solution is to decrease the time step, as we shall see in Sect. 5.

The training period is terminated when the CPT trajectory starts to deviate from the truth beyond a given pre-specified threshold. After training an optimal trajectory is obtained that is produced by a combination of different imperfect models (Fig. 4). Next we count how often during training each model has produced the best prediction of a particular component of the state vector. This frequency of occurrences is used to compute weights $\mathbf{W}$ for the corresponding time-derivative of the state

10    vector. This superposition of weighted imperfect models forms a weighted supermodel, as expressed in the example of Eq. 2.

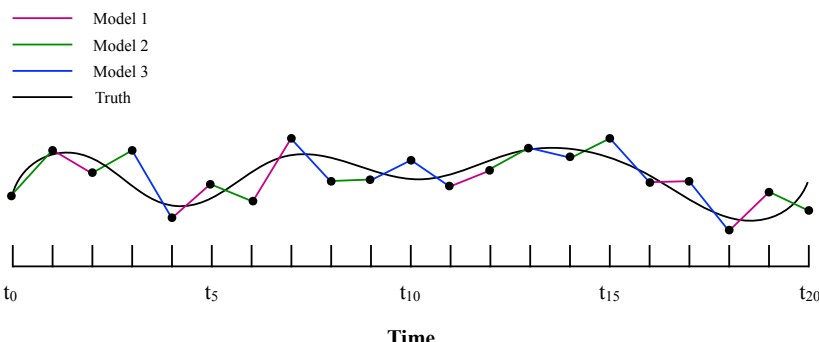

**Figure 4.** CPT trajectory after a training period of 20 time steps. Model 1 is used for 6 out of 20 time steps, hence model 1 will get a weight of 0.3.

## 4.2 Synchronization based learning

For the training of a supermodel based on synchronization a learning rule (the synch rule) is used that updates the weights such that synchronization errors between truth and supermodel are minimized. In contrast to CPT learning, initial values for the weights need to be chosen and the weights are updated during training. Under certain conditions the supermodel will fall

5 into synchronized motion with the truth as the weights are updated and the supermodel is nudged to the truth (black arrows in Fig. 5).

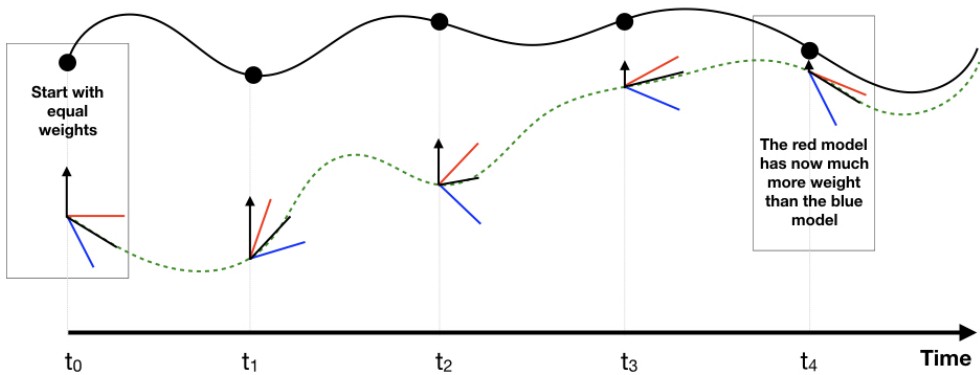

**Figure 5.** At each observation (dots) of the truth (continuous black line) the weights of the imperfect models (red, blue) are updated which gives a new supermodel solution (green dotted line). The black arrows indicate the nudging to the truth.

The synch rule for the weights is an application of the general synchronization based parameter estimation approach suggested in (Duane et al., 2007). Recently the synch rule was applied to train the connections in a connected SPEEDO supermodel





(Selten et al., 2017). We follow a similar strategy and implement the synch rule to train the weights of a weighted SPEEDO supermodel.

In the context of two dynamical systems that differ in parameter values only, the general synch rule for parameter estimation is given by

$$\dot{\mathbf{x}} = \mathbf{f}(\mathbf{x}; \mathbf{p}) \tag{6a}$$

$$\dot{\mathbf{y}} = \mathbf{f}(\mathbf{y}; \mathbf{q}) - \mathbf{K}(\mathbf{y} - \mathbf{x}) \tag{6b}$$

$$\dot{q}_j = -\delta_j \sum_i e_i \frac{\partial f_i(\mathbf{y}, \mathbf{q})}{\partial q_j}, \tag{6c}$$

where $\mathbf{p}$ and $\mathbf{q}$ are vectors of parameters. $\mathbf{K}(\mathbf{y} - \mathbf{x})$ is a connecting term between the two systems that nudges $\mathbf{y}$ towards $\mathbf{x}$. $\mathbf{K}$ is a diagonal matrix of nudging coefficients, $\mathbf{K} = \mathrm{diag}(\mathbf{k})$. Suppose the two systems (6a) and (6b) synchronize if $\mathbf{p} = \mathbf{q}$, that is, as $t \to \infty$, $\mathbf{y}(t) \to \mathbf{x}(t)$. We further assume that the parameters appear only linearly in the model equations. Then it can be proven that, using the learning rule (6c), even if the two systems are not identical, $\mathbf{p} \neq \mathbf{q}$, the systems will still synchronize and the parameters will become equal, $\mathbf{q}(t) \to \mathbf{p}$ as $t \to \infty$. Here $q_j$ denote the parameter values, with $j$ indexing the elements of the parameter vector. Furthermore $e_i = y_i - x_i$ denotes the synchronization error at the current time step with $i$ indexing the elements of the state vector and $\delta_j$ an adjustable rate of learning scaling factor. Every time step the update $\dot{q}_j$ for the weight $q_j$ is calculated.

In training a supermodel, we assume that the truth can be described by a weighted dynamical combination of imperfect models with the weights as adjustable parameters. In this case the function $\mathbf{f}$ corresponds to the supermodel, $\mathbf{q}$ corresponds to the weights of the supermodel, $\mathbf{x}$ denotes the truth and $\mathbf{y}$ the supermodel solution. In our SPEEDO case it is not exactly true that the truth can be described as a weighted superposition of imperfect models since the perturbed parameters do not appear linearly in the equations, but apparently the approximation is close enough for the learning rule to work well.

Integration of the synch rule implies that as long as the time-series of the synchronization error $e_i$ and the effect of the parameter on the imperfect model evolution $\frac{\partial f_i(\mathbf{y}, \mathbf{q})}{\partial q_j}$ are correlated, the parameter will be updated. For instance, when a parameter update systematically enhances warming in the model when the model is colder than the truth and the same holds when the parameter update systematically cools the model when it is too warm, then the updated parameter will decrease the synchronization error between the model and truth over time. When this correlation vanishes, hence there is no systematic relation anymore between updating the parameter and the state of the model, then systematic updates cease. When perfect synchronization is reached, hence $e_i = 0$, then naturally updates also stop.

## 5 Training in SPEEDO

In training the SPEEDO supermodel we regard the atmospheric model with standard parameter values as truth whereas imperfect atmospheric models are created by perturbing those parameter values. Figure 6 depicts the configuration during training. All atmosphere models are independently coupled to the same ocean and the land model. They each calculate their own water, heat and momentum fluxes and receive the information from the ocean and the land model from the truth only.



During training the truth and imperfect models all share their states. In case of CPT, this state information is used by each imperfect model to check which model is closest to the truth and continue the integration from that state. In case of the synch rule this state information is used to calculate the nudging terms.

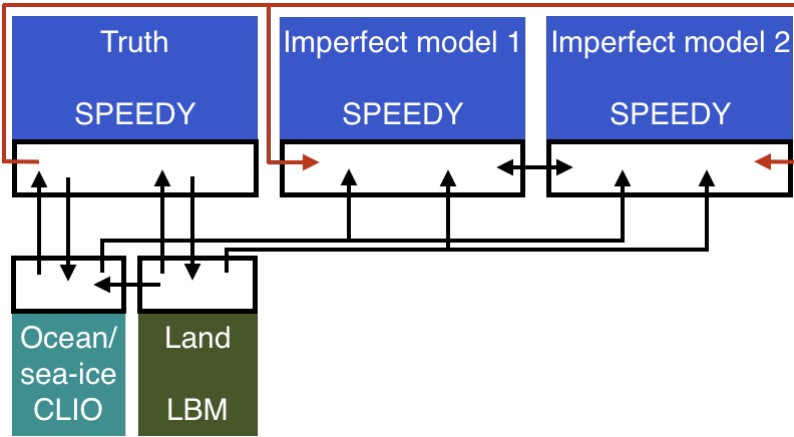

**Figure 6.** Schematic representation of the SPEEDO system during training (Selten et al., 2017).

Application of the synch rule to a weighted SPEEDO supermodel of two imperfect models implies integration of the following set of equations

$$\dot{\mathbf{a}}_0 = \mathbf{f}^a(\mathbf{a}_0; \mathbf{p}_0^a) + \mathbf{g}^a(\mathbf{e}_0^h, \mathbf{e}_0^w, \mathbf{e}_0^m) \tag{7a}$$

$$\dot{\mathbf{a}}_1 = \mathbf{f}^a(\mathbf{a}_s; \mathbf{p}_1^a) + \mathbf{g}^a(\mathbf{e}_1^h, \mathbf{e}_1^w, \mathbf{e}_1^m) - \mathbf{K}(\mathbf{a}_s - \mathbf{a}_0) \tag{7b}$$

$$\dot{\mathbf{a}}_2 = \mathbf{f}^a(\mathbf{a}_s; \mathbf{p}_2^a) + \mathbf{g}^a(\mathbf{e}_2^h, \mathbf{e}_2^w, \mathbf{e}_2^m) - \mathbf{K}(\mathbf{a}_s - \mathbf{a}_0) \tag{7c}$$

$$\dot{\mathbf{o}} = \mathbf{f}^o(\mathbf{o}; \mathbf{p}^o) + \mathbf{g}^o(\mathcal{P}^o\mathbf{e}_0^h, \mathcal{P}^o\mathbf{e}_0^w, \mathcal{P}^o\mathbf{e}_0^m, \mathcal{P}^o\mathbf{r}) \tag{7d}$$

$$\dot{\mathbf{l}} = \mathbf{f}^l(\mathbf{l}; \mathbf{p}^l) + \mathbf{g}^l(\mathcal{P}^l\mathbf{e}_0^h, \mathcal{P}^l\mathbf{e}_0^w, \mathbf{r}) \tag{7e}$$

$$\dot{\mathbf{a}}_s = \mathbf{W}_1\dot{\mathbf{a}}_1 + \mathbf{W}_2\dot{\mathbf{a}}_2 \tag{7f}$$

$$\dot{W}_{i,j} = -\delta_j(a_{s,j} - a_{0,j})\dot{a}_{i,j}, \tag{7g}$$

where index 0 refers to the truth and $W_{i,j}$ refers to the weight of model $i$ and state vector element $j$. During training we choose a uniform nudging strength corresponding to a 24 hours time scale, as motivated by Selten et al. (2017). They showed that for this value of $\mathbf{K}$, two connected identical SPEEDO models (perfect model scenario) almost perfectly synchronize with very small synchronization errors in temperatures of order 0.01 degree Celsius. Imperfect models on the other hand have synchronization errors with respect to the truth that are usually 10 times larger, therefore with the synch rule errors cannot be reduced below 10 times the synchronization error in a perfect model configuration.





## 5.1 Construction of imperfect models

In order to be able to compare results of the weighted supermodels of this study to the connected supermodels in Selten et al. (2017), we choose the same parameter values for the imperfect models. These parameters are the convection relaxation timescale, the relative humidity threshold and the momentum diffusion timescale. The reason to perturb these parameters is

because the uncertainty in climate models mostly lies in the parameterization of clouds and convection, and perturbing these parameters in the SPEEDO model results in a spread in the simulated climate that characterizes this uncertainty. The parameters are listed in Table 1 where model 1 and model 2 correspond to the imperfect models of Selten et al. (2017). The impact of the parameter perturbations on the climate (i.e. long term behavior) of the models is assessed on the basis of 40 year simulations initiated on January 1st of model year 2001 of a long control simulation as in Selten et al. (2017). Figure 7 shows the global

mean time series from the six imperfect models of Table 1 plus the truth, for different variables. From the figure it appears evident how the imperfect models all drift away from the truth giving raise to biases. For example: the global mean temperature of imperfect model 1 rises about 1.5 degrees within a couple of decades, whereas model 2 cools around 0.5 degrees. These global mean temperature biases are comparable to the biases of state-of-the-art global climate models compared to real world observations (IPCC, 2013). The first supermodel that we will train will consist of a weighted superposition of model 1 and 2.

The second supermodel will consist of a weighted superposition of models 1,3,4 and 5. The parameter values of these models are chosen such that they form a so-called convex hull around the true parameter values (see Schevenhoven and Selten (2017) for a discussion on the convex hull principle). Note that we use only two perturbed values for each parameter, the imperfect models differ only in the combination of these values, such that in the four-model supermodel a convex hull is formed. This implies that, provided the model functional dependence on the parameter is linear, the true parameter values can be obtained

as a linear combination with positive coefficients/weights of the four parameter values of the imperfect models. While these conditions do not perfectly hold in this case, we expect that we can create a weighted supermodel based on these four models that will be close to the truth. All of these four models overestimate the global mean temperature and precipitation (Table 2). Therefore simply taking the MME mean with positive weights will not produce a climatology closer to the truth. We expect however that, based on the convex hull principle, the weighted supermodel will nevertheless be able to produce a climatology

that is closer to the truth.

The third supermodel consists of a weighted superposition of models 1 and 6. In this case, both imperfect models have parameter values that are smaller than the corresponding true values. A weighted superposition with positive weights does not correspond to a model with parameter values that are closer to the truth. Note that both models overestimate the average temperature and precipitation (Table 2) hence taking the MME mean with positive weights also does not produce a climatology

closer to the truth. In this case we will explore whether a weighted supermodel with negative weights can be trained in order to improve the climatology and short-term forecasts.



**Table 1.** Parameter values of perfect and imperfect models.

| model | convection relaxation timescale | relative humidity threshold | momentum diffusion timescale |
|---|---|---|---|
| perfect | 6 hours | 0.9 | 24 hours |
| model 1 | 4 hours | 0.85 | 18 hours |
| model 2 | 8 hours | 0.95 | 30 hours |
| model 3 | 4 hours | 0.95 | 30 hours |
| model 4 | 8 hours | 0.95 | 18 hours |
| model 5 | 8 hours | 0.85 | 30 hours |
| model 6 | 3 hours | 0.75 | 14 hours |

**Table 2.** Global mean average difference between the imperfect models and the perfect model, calculated over the last 30 years of the simulation.

| model | temperature $[C°]$ | precipitation $[mm/day]$ | wind at 200 hPa $[m/s]$ | wind at 850 hPa $[m/s]$ | solar surface radiation $[W/m^2]$ | cloudcover $[\%]$ |
|---|---|---|---|---|---|---|
| model 1 | 1.37 | 0.11 | 1.04 | 0.07 | 2.06 | -1.59 |
| model 2 | -0.38 | -0.04 | -0.31 | -0.03 | -1.13 | 0.87 |
| model 3 | 0.99 | 0.10 | 1.14 | 0.06 | 1.21 | -1.03 |
| model 4 | 0.45 | 0.04 | -0.04 | -0.01 | -0.20 | 0.10 |
| model 5 | 0.86 | 0.08 | 0.72 | -0.01 | -0.19 | -0.12 |
| model 6 | 3.20 | 0.26 | 2.25 | 0.03 | 3.95 | -3.37 |

## 5.2 Global weights

For both CPT and the synch rule we choose to work with global weights, which means that for each meteorological variable we use the same weight at every grid point. In principle, one could allow different weights per each grid point but it could induce dynamic imbalances that pull the model away from its attractor. The model's reaction is then to restore the dynamical balances and return to its own attractor (Pecora and Carroll, 1990). In SPEEDO this leads to the generation of fast gravity waves and fast convective adjustments. An adequately small time step is required in order to prevent numerical instabilities. We choose instead to use global weights in order to limit the computational time.



## 5.3 Exchange of state information

The SPEEDO model has five prognostic variables: temperature, vorticity, divergence, specific humidity and surface pressure (T,VOR,DIV,TR,PS). Best results were obtained by limiting the weighted averaging of state information to temperature, vorticity and divergence only. We suspect that exchanging specific humidity and surface pressure leads to imbalances and fast spurious adjustments that deteriorate the supermodel solution. We found that a perfect SPEEDY atmosphere only fully synchronizes with the truth when at least temperature, vorticity and divergence are nudged to the truth (not shown). Therefore in a weighted supermodel at least these variables need to be exchanged.

## 5.4 Required time step

We found that smaller time steps were required during CPT training as compared to standard integrations. Gravity waves induced by the state replacement during training require a smaller time step in order to prevent numerical instabilities. We found that a 15 minute time step was sufficient with our choice of imperfect models, half the time step of the standard integration.

## 5.5 Initialization of the weights for the synch rule

In CPT training, the sum of the weights is normalized to one. In the application of the synch rule on the other hand the sum of the weights is not explicitly constrained. One can start from zero weights and let the synch rule find the optimal set of weights. Initializing weights with sum larger than one easily leads to numerical instabilities because the weighted mean state becomes more energetic. Imposing the constraint of the sum of weights being one during the training also led to numerical instabilities. We chose to initialize with equal weights that sum to one.

## 5.6 Rate of learning in the synch rule

The synch rule contains an adjustable rate of learning scaling factor $\delta_j$, with $j$ the index of the state vector. A large rate of learning is desirable since it leads to faster convergence and shorter training periods. However, the parameters should vary on a slower time-scale than the dynamical variables and this provides an upper bound for the value of $\delta_j$. Furthermore it turns out that if $\delta_j$ is too large, the sum of the weights can become greater than one which easily leads to numerical instabilities. The size of $\delta_j$ in the synch rule depends on the variable that is being exchanged and was determined by trial and error during the training experiments. The largest values for $\delta_j$ that resulted in converged weights were in the order of $10^7$ for divergence and vorticity and $10^{-4}$ for temperature. With these scaling factors about similar rates of learning were achieved for the different variables. This makes sense since the state values for divergence and vorticity are much smaller than for temperature, so the product of $\delta_j$ and the state values in the synch rule is of the same order of magnitude.

## 5.7 Weights for the heat, water and momentum fluxes

In the experimental setup during training, we assume a perfect ocean and land model which receive fluxes from the perfect atmosphere. However, in the supermodel setup, perfect fluxes are not available and we use a weighted combination of the





fluxes from both imperfect models instead. In the connected supermodel of Selten et al. (2017) the fluxes are averaged using equal weights. In this paper we further optimize the weights for the fluxes, because we found they have a big influence on the supermodel's performance. In particular, we selected weights given by the average of the three weights for the prognostic variables. To check whether this choice was optimal, we used a least squares minimization method in order to optimize the

weights for the fluxes. During one year of training, the fluxes from the perfect and imperfect models were saved at every time step. The weights were determined by a least squares fit of a weighted sum of the imperfect fluxes to the perfect fluxes. The flux weights obtained from the minimization method did differ slightly per flux (heat, water or momentum flux), but the average weights were close to the average of the weights for the prognostic variables.

## 6 Results

We describe the learning results and the forecast short- and long-term capabilities of the three supermodel configurations separately.

### 6.1 Supermodels based on two imperfect models

We first trained a weighted supermodel based on imperfect model 1 and 2 (see Table 1) applying both CPT and the synch rule. As a benchmark we compare the quality of the weighted supermodel after training with the connected SPEEDO supermodel

of Selten et al. (2017). This supermodel is based on the same imperfect models and was trained by the synch rule.

Ideally both CPT and the synch rule should produce converged weights, i.e. weights that remain stable if the training period is extended. The required length of the training period for the convergence of the two methods turns out to be very different. For CPT, a training period as short as a couple of days produces converged weights, whereas for the synch rule it takes about a year. Note that we limit the CPT training period to a week, as the CPT trajectory starts to deviate significantly from the truth

after approximately 10 days. The reason that CPT diverges from the truth is because we have a limited ensemble size. With non-linear processes causing rapid error growth the truth soon falls outside the limited ensemble. The problem is exacerbated by replacing a model state with state variables mixed from different models which introduces imbalances that cause additional error growth.

In order to check the difference between the CPT weights during a year, the CPT method is applied for each week during

one year. After each week, the values for all prognostic variables are reset to the truth, and the procedure is repeated. Figure 7a shows the values of the weights during training. The weights for both temperature and vorticity remain fairly constant. The weights for divergence vary within $0.04$ of a mean value. For the final supermodel weights we just take the average over the whole year (Table 3).

Using the synch rule, weights for temperature and vorticity converge within the first couple of weeks, whereas for divergence

the weights cannot be learned faster than within a year in order to avoid numerical instabilities (see Fig. 7b) . When using the synch rule, the weights converge to similar values as compared to the CPT training (Table 3). Converged values of both





methods are within 0.05. Whether these small differences matter for climate and weather forecasts will be assessed in the next two sections. Although not imposed, the training yields sum of weights equal to one as an optimal solution.

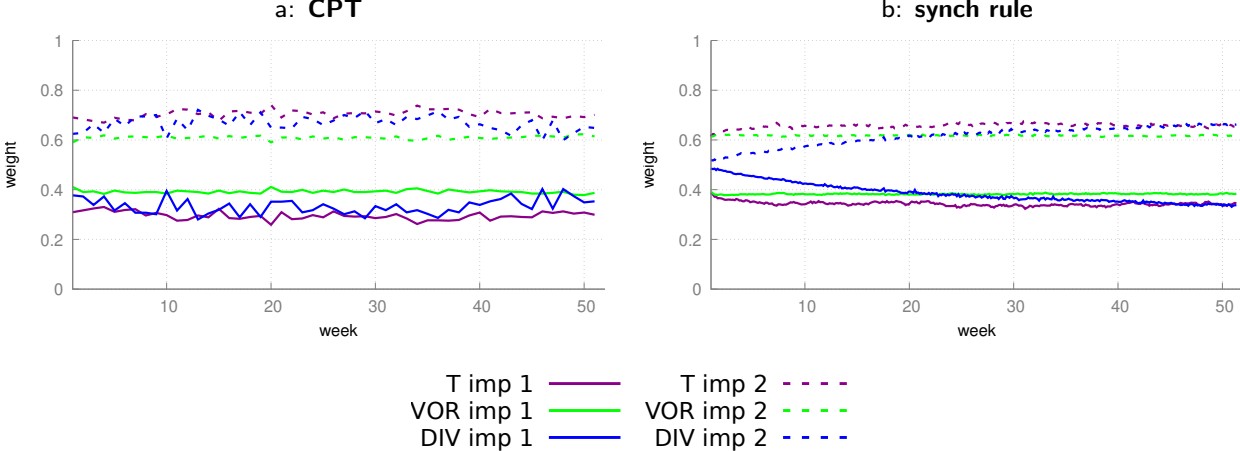

**Figure 7.** Calculation of weights for a supermodel constructed from two imperfect models using two different training schemes. (a) CPT weights calculated during a training period of one week estimated for each week of a year. (b) Weights for the synch rule during a training period of one year.

**Table 3.** Weights for the supermodel trained by CPT and the synch rule. Between brackets the standard deviation over the year (CPT) or the standard deviation over the last 10 weeks of training (synch rule) is given.

| model | method | T | VOR | DIV |
|---|---|---|---|---|
| model 1 | **CPT** | 0.30 | 0.39 | 0.35 |
| | | (0.016) | (0.007) | (0.031) |
| model 2 | | 0.70 | 0.61 | 0.65 |
| | | (0.016) | (0.007) | (0.031) |
| model 1 | **synch rule** | 0.35 | 0.38 | 0.34 |
| | | (0.0043) | (0.0018) | (0.0052) |
| model 2 | | 0.65 | 0.62 | 0.66 |
| | | (0.0043) | (0.0018) | (0.0053) |

### 6.1.1 Climate measures

The imperfect models and the supermodel are integrated for 40 years in time, starting from January 1st of model year 2001. The climatology is defined as the average over years 11-40. The error in the climatology is defined as the root of the global mean squared difference (RMSE) between the model and the truth. In addition, the perfect model is integrated for 40 years from a





slightly perturbed initial condition, in order to obtain an estimate of the sampling error, i.e. to estimate the representativeness of the errors of the different models. Global mean time-series for surface air temperature, precipitation, surface solar radiation and cloud cover for the different models show that both weighted supermodels behave very similar and remain close to the perfect model (Fig. 8). The errors in the climatologies of the various fields of both supermodels are much reduced as compared to

5   both imperfect models and are indistinguishable from the statistical sampling error of the perfect model. Both training methods succeed in greatly improving the simulation of the climate. Compared to the trained connected supermodels of Selten et al. (2017), the weighted supermodels have reduced climatological errors (see Fig. 15). Training of a connected supermodel by the synch rule on the other hand is more efficient as faster learning rates could be used, leading to convergence within two weeks of training.

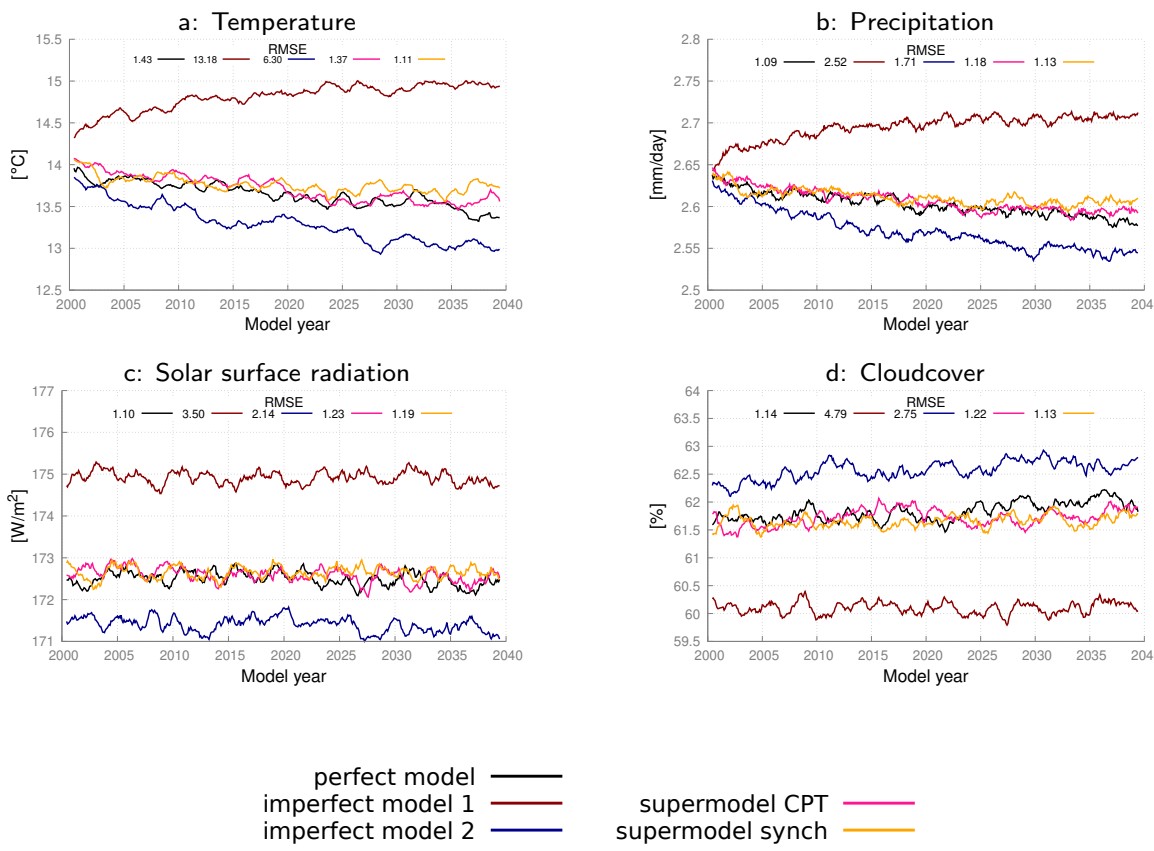

**Figure 8.** Global mean time-series for the perfect model, the imperfect models and the two supermodels trained by CPT and the synch rule. The normalized RMSE in the climatology of model years 2011-2040 with respect to the climatology of the truth is given in each panel. The normalization is such that the expected value of the perfect model error is one.





A spatial characterization of the performance of the supermodel in simulating the climatology of the zonal wind at 200 hPa is given in Fig. 9. Clearly, both supermodels outperform the imperfect models and their local errors are of similar magnitude as the sampling error of the perfect model. We computed an optimal weighted average of the climatology of both imperfect models (optimal in the sense that the RMSE in the climatology is minimized) as in Selten et al. (2017). This MME mean

5 climatology (Fig. 9f) has errors in the same order of magnitude as both trained weighted supermodels due to fact that the imperfect model errors are near-mirror images of each other.

In the context of simpler models, Schevenhoven and Selten (2017) noted that CPT training of a couple of days duration was sufficient to reduce climatological errors substantially and this result carries over to the complex SPEEDO model used here. This notion that errors in fast processes contribute substantially to errors in the long-term mean state is also supported by other studies, for example by Rodwell and Palmer (2007). Since the climatological errors are reduced, we expect the trained supermodels to produce better short-term forecasts as compared to the imperfect models.

### 6.1.2 Forecast quality

In order to assess the quality of short-term forecasts, we initialized the various models from slightly perturbed states of the truth

and integrated the models for two weeks. We selected 25 initial states, two weeks apart, starting January 1st, so the forecasts cover almost one year. The quality of the forecast is measured by the RMSE in the global surface air temperature forecast, averaged over the 25 forecasts and is shown in Fig. 10. In these forecasts, the atmosphere models are forced by the ocean and land conditions of the truth, this is to exclude error growth related to the coupled interactions. As expected, the RMSE in surface air temperature of the perfect model is the one growing the slowest, and it is still as small as about 0.3 degree at day

14. On the other hand, the forecast errors of both imperfect models is 0.3 degrees around day 3 and grow to over 3 degrees at day 14. Both trained weighted supermodels reach 0.3 degrees around day 8 and over 1 degree at day 14. For comparison we computed the forecast error of the weighted mean forecast of both imperfect models using the same weights as used in Fig. 9 in the calculation of the optimal climatology. This MME mean forecast has smaller forecast errors than the imperfect models, yet both supermodels are clearly superior.

### 6.2 Supermodels based on four imperfect models forming a convex hull

As explained in Sect. 5.1, the parameter perturbations of models 1,3,4 and 5 form a convex hull around the true parameter values (Table 1). We therefore expect to be able to create a weighted supermodel based on these four models that will be close to the truth, despite the fact that all four have a warmer climatology than the truth (see Table 2). The weights are trained using both CPT and the synch rule in the same way as in the previous case with two imperfect models and are shown in Fig. 11; nevertheless given that the supermodels are now based on four imperfect models, the number of weights is doubled. Again the weights during CPT training vary from week to week within 0.05 and converge within a year using the synch rule. Weights

5 for vorticity turn out to be a special case since the change in vorticity as calculated by imperfect models 1 and 3 is equal to respectively model 4 and 5. The reason is that only the perturbation in the momentum diffusion timescale affects the vorticity change and model 1 and 3 have the same diffusion timescale as in model 4 and 5 respectively. Therefore their weights are

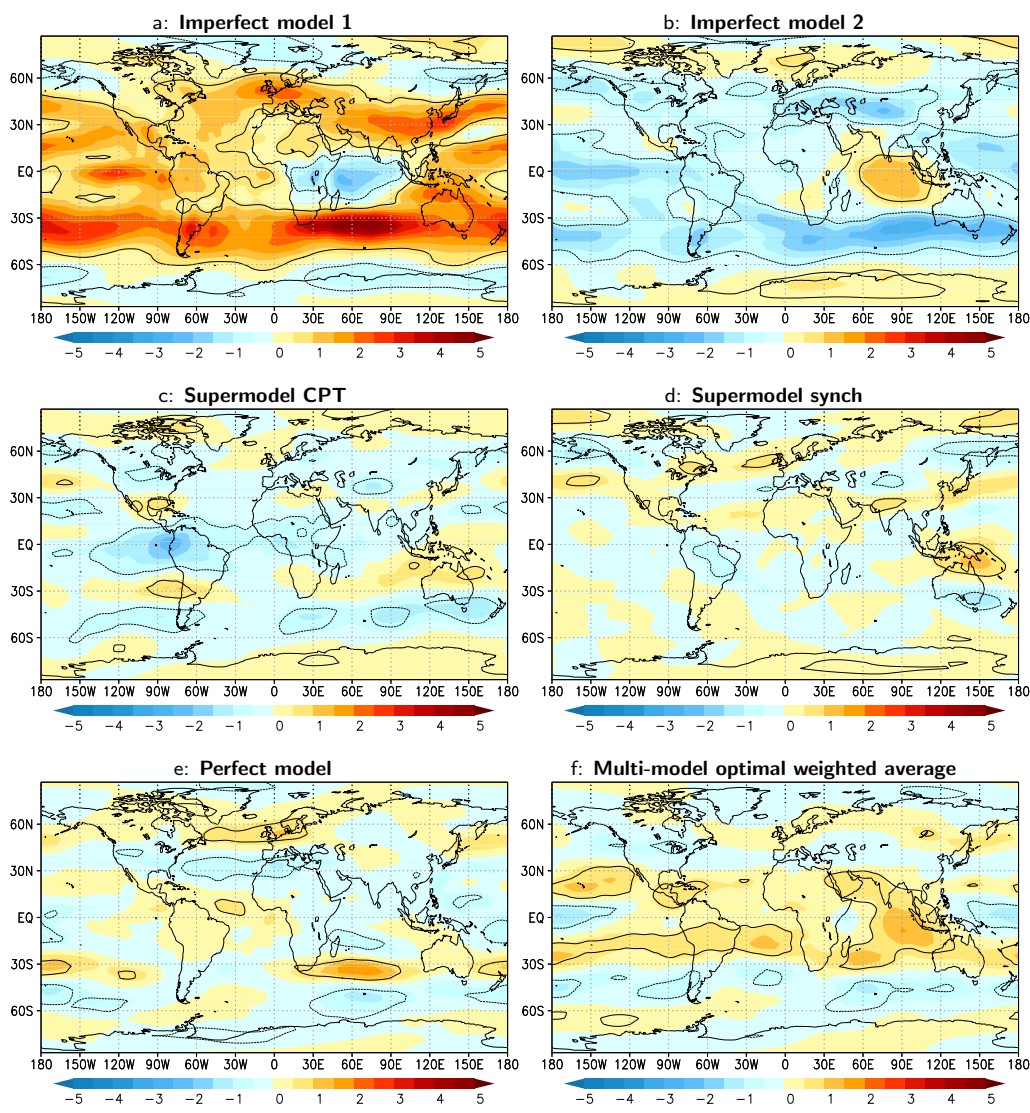

**Figure 9.** Difference in the zonal wind at 200 hPa averaged over model years 2011-2040 for the various models with respect to the truth. Contours denote areas where the difference is larger than the sampling error at 95% confidence (solid for positive difference, dotted for negative). Positive values imply stronger mean winds blowing eastward. Units: m/s.





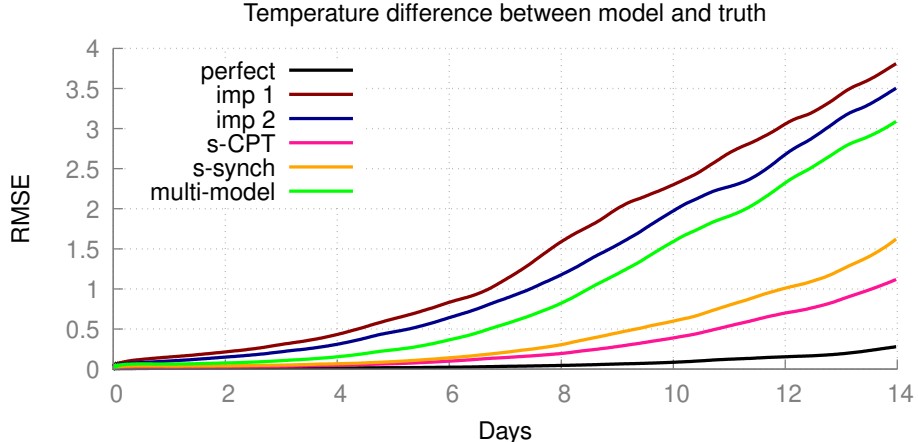

**Figure 10.** Forecast quality as measured by the root-mean-squared error (RMSE) of the truth and a model with a perturbed initial condition. The control is the difference between the perfect model and the perfect model with a perturbed initial condition.

equal. Table 4 denotes the final supermodel weights, where for vorticity the weight is equally distributed over models 1 and 4 and model 3 and 5.

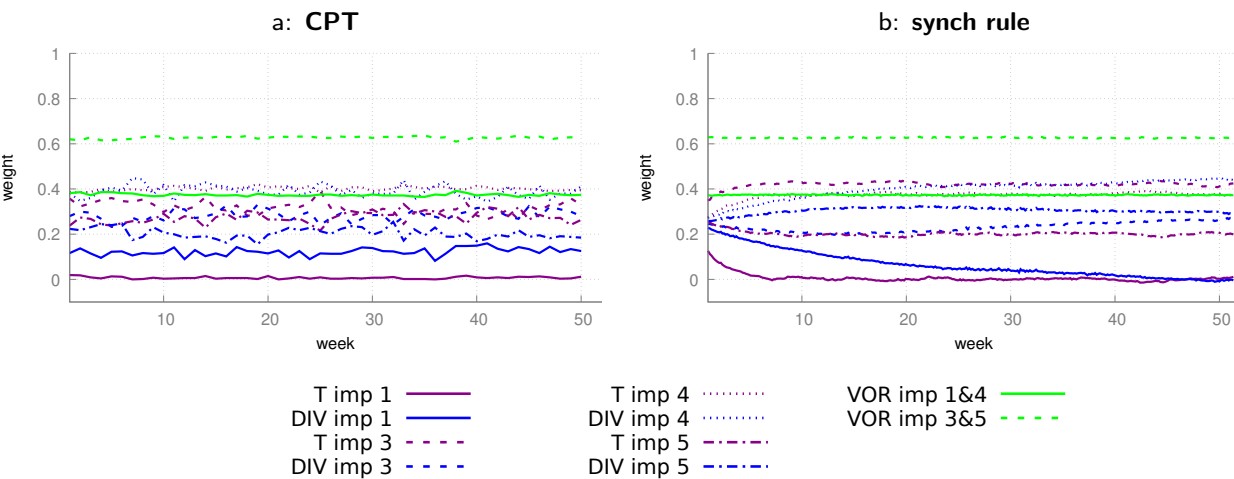

**Figure 11.** CPT weights calculated during a training period of one week for one year (left) and the weights for the synch rule for a training period of one year (right) with four imperfect models.

10  The value of the weights for vorticity trained by the synch rule are very close to the values obtained by CPT training. This is not the case for temperature and divergence. For temperature, CPT puts $10\%$ less weight on imperfect model 3 compared to





**Table 4.** Weights for the supermodel trained by CPT and the synch rule. Between brackets the standard deviation over the year (CPT) or the standard deviation over the last 10 weeks of training (synch rule) is given.

| model | method | T | VOR | DIV |
|-------|--------|-----|-----|-----|
| model 1 | **CPT** | 0.01 | 0.19 | 0.12 |
|  |  | (0.005) | (0.022) | (0.017) |
| model 3 |  | 0.33 | 0.31 | 0.28 |
|  |  | (0.030) | (0.037) | (0.024) |
| model 4 |  | 0.40 | 0.19 | 0.41 |
|  |  | (0.009) | (0.022) | (0.028) |
| model 5 |  | 0.26 | 0.31 | 0.19 |
|  |  | (0.026) | (0.040) | (0.023) |
| model 1 | **synch rule** | 0.01 | 0.19 | 0.00 |
|  |  | (0.0070) | (0.0007) | (0.0064) |
| model 3 |  | 0.42 | 0.31 | 0.27 |
|  |  | (0.0072) | (0.0007) | (0.0047) |
| model 4 |  | 0.37 | 0.19 | 0.44 |
|  |  | (0.0065) | (0.0007) | (0.0044) |
| model 5 |  | 0.20 | 0.31 | 0.29 |
|  |  | (0.0062) | (0.0007) | (0.0029) |

the synch rule and a $10\%$ stronger weight on model 1 for divergence. The synch rule puts (almost) zero weight on imperfect model 1. This is because imperfect model 1 calculates exactly the same vorticity change as imperfect model 4 hence the synch rule suggests that imperfect model 1 has no added value in the weighted supermodel. Again, the synch rule training yields sum of weights equal to one as an optimal solution. Using these weights we will compare the climatology and forecast skill of both supermodels.

### 6.2.1 Climate measures and forecast quality

We repeated similar climate integrations as in case of the supermodels based on two imperfect models and assessed the climatological errors. By comparing the 40-year time-series of global mean values in Fig. 12, both supermodels remain close to the perfect model and are clearly superior to all the imperfect models. Despite all imperfect models become too warm and precipitate too much on the global scale, the supermodels achieve to balance model deficiencies and produce climate simulations that are close to the truth. Inspection of the RMSE of the 30 year mean fields in the different figure panels indicates that for temperature the supermodel with weights from the CPT training is substantially better than the supermodel with weights from the synch rule. Recall that while imperfect model 1 almost does not contribute to the supermodel with the weights from



**Figure 12.** Global mean time-series for the truth, the perfect model, the imperfect models and the two supermodels trained by CPT and the synch rule. Included is the RMSE of the model years 2011-2040 with respect to the truth. The normalized root-mean-squared error (RMSE) in the climatology of model years 2011-2040 with respect to the climatology of the truth is given in each panel. The normalization is such that the expected value of the perfect model error is one.





the synch rule, it does so for the supermodel with the weights from the CPT training. Although imperfect model 1 has larger
10   climatological errors than the other imperfect models (Fig. 12), it nevertheless improves the quality of the CPT supermodel.

This experiment demonstrates the potential of supermodels to mitigate common errors, and thereby clearly outperform the standard MME-approach. Since all imperfect models overestimate the global average temperature and simulate too much precipitation, a standard weighted MME-approach results in a climatological forecast worse than the best imperfect model. In the case that the imperfect parameters form a convex hull around the true parameter values, we may expect that a supermodel
15   can be constructed with a climatology much closer to the truth as compared to the best imperfect model. In the case that the imperfect models do not form a convex hull around the true parameter values, allowing negative weights in the weighted supermodel might still improve the climatology and forecast skill. This will be explored in the next section.

We repeated the same forecast experiment as in the case of the supermodel based on two imperfect models. Also in this case the supermodels have forecast errors that are substantially reduced as compared to the imperfect models, up to a factor of three
smaller (not shown). Both supermodels have comparable forecast skill in this measure.

## 6.3   Negative weights

The CPT training method only produces positive weights, since the weights are defined as being equal to the frequency that the solution of a particular model is closest to the truth during the training period. The synch rule training on the other hand does not impose any constraint on the weights. The weights came out positive due to the convex-hull principle: the imperfect models
considered so far surrounded the truth and with positive weights the effect of the true parameter values can be approximated. But in case the imperfect models have parameter values that are all smaller or larger than the truth, only by allowing negative weights one can construct a linear superposition of imperfect models that is closer to the truth. To test if such a supermodel with negative weights indeed shows the desired physical behavior and to test if we can obtain such a model with the synch rule, we construct a weighted supermodel based on two imperfect models (models 1 and 6) with parameter values on the same side
of the true parameter values (Table 1).

After a training period of one year using the synch rule, stable weights are obtained, which indicates that at least a local minimum is reached. And as expected, the training produces negative weights (Table 5). In contrast to the previous experiments however, the weights for temperature, divergence and vorticity are quite different. The weights for divergence are positive and do not substantially differ from the weights of the previous experiments. The weights for temperature and vorticity are negative
for one of the imperfect models and larger than one for the other such that the sum is again close to one.

Stable climate simulations turn out to be possible with a weighted supermodel using negative weights. The climatology of the supermodel has improved significantly compared to both imperfect models as displayed in Table. 6. Global mean values of the various fields are closer to the truth, despite the fact that the global mean climatological errors of both imperfect models have the same sign. Also local model errors largely have the same sign but are smallest for the supermodel as shown in Fig. 13 for the zonal wind at 200 hPa. Nevertheless, despite the improvement, substantial errors still remain in the supermodel solution.





**Table 5.** Weights for the supermodel trained by the synch rule. Between brackets the standard deviation over the last 10 weeks of training is given.

| model | T | VOR | DIV |
|---|---|---|---|
| model 1 | 1.30 | 2.00 | 0.40 |
| | (0.016) | (0.011) | (0.010) |
| model 2 | -0.30 | -1.00 | 0.60 |
| | (0.016) | (0.010) | (0.009) |

**Table 6.** Global mean average difference with the perfect model, calculated over the last 30 years of the simulation.

| model | temperature $[C°]$ | precipitation $[mm/day]$ | solar surface radiation $[W/m^2]$ | cloudcover $[\%]$ |
|---|---|---|---|---|
| model 1 | 1.37 | 0.11 | 2.06 | -1.59 |
| model 6 | 3.20 | 0.26 | 3.95 | -3.37 |
| supermodel | 0.64 | 0.06 | 1.68 | -1.16 |

The forecast errors are evaluated in a similar fashion as in the previous cases and shown in Fig. 14. Although there is a significant improvement in quality for the supermodel as compared to the imperfect models, the forecast error is still quite large. Closer correspondence to the truth can only be expected if all prognostic variables are exchanged, hence also specific humidity and surface pressure, and if the perturbed parameters appear linear in the equations. Both conditions are not fulfilled in this case.

### 6.4 Summary of supermodel climate errors

We conclude this section with a summary of the climatological errors of the weighted supermodels of this study and the connected supermodel of Selten et al. (2017) in Fig. 15. The climatological errors of the weighted supermodels of this study based on two imperfect models are of the order of the sampling error of the perfect model, whereas the connected supermodel based on the same two imperfect models has substantially larger errors. Also the weighted supermodel based on the four imperfect models trained by CPT is indistinguishable from the truth with respect to its climatological errors, whereas the synch rule trained weighted supermodel has substantially larger errors. These results suggest that CPT training might yield more robust results. Largest climatological errors remain for the supermodel with negative weights.

### 7 Discussion and conclusions

We have demonstrated the potential of weighted supermodeling to improve weather and climate predictions using the global coupled atmosphere-ocean-land model SPEEDO in the presence of parametric error. Weighted supermodels are constructed



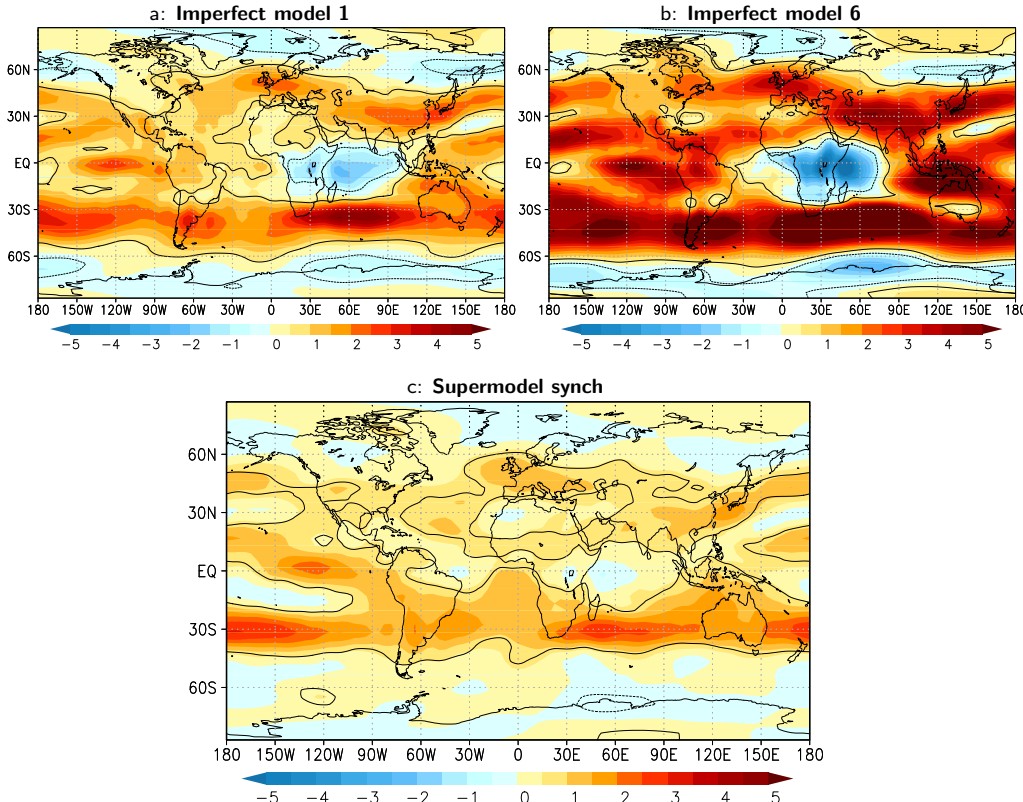

**Figure 13.** Difference in the east-west component of the wind at the 200 hPa pressure level averaged over model years 2011-2040 for the various models with respect to the truth. Contours denote areas where the difference is larger than the sampling error at 95% confidence (solid for positive difference, dotted for negative). Positive values imply stronger mean winds blowing eastward. Units: m/s.

based on SPEEDO with perturbed parameters. The perturbations are chosen such that the spread in imperfect models reflects the uncertainty in climate models realistically. The weights are trained using data from the perfect model (i.e. our reference simulated truth) using two different training schemes having low computational cost. The first method is based on Cross Pollination in Time (CPT), where different model trajectories are "crossed" in order to create a larger ensemble of possible trajectories. The second method is a synchronization based learning rule (synch rule), which adapts the weights of the different imperfect models during training such that the supermodel synchronizes with the perfect model.

Both training methods yield supermodels that outperform the individual imperfect models, in short-term forecasts as well as in long-term climate simulations. CPT training required shorter training periods (one week as opposed to a year for the synch rule), but both are much more efficient than cost function based approaches that are known to require many climate simulations in an iterative process to reach convergence on optimal weights (van den Berge et al., 2011; Shen et al., 2016). An advantage

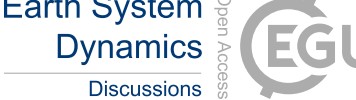

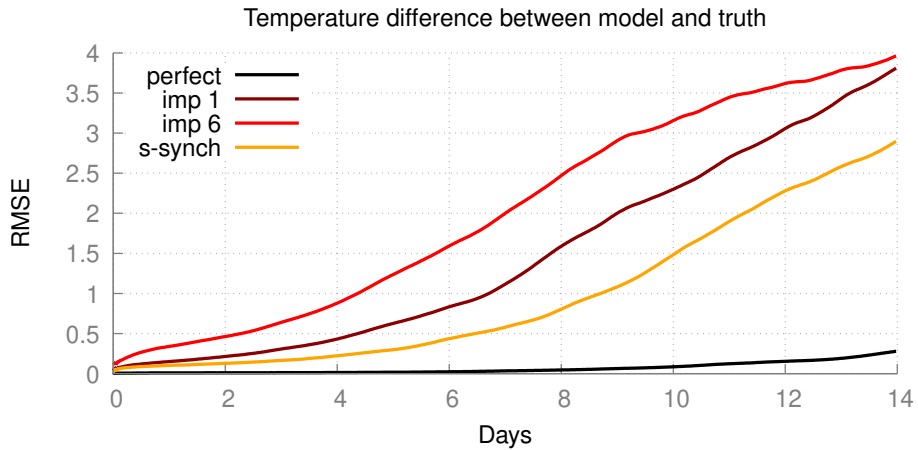

**Figure 14.** Forecast quality as measured by the root-mean-squared error (RMSE) of the truth and a model with a perturbed initial condition. The control is the difference between the perfect model and the perfect model with a perturbed initial condition.

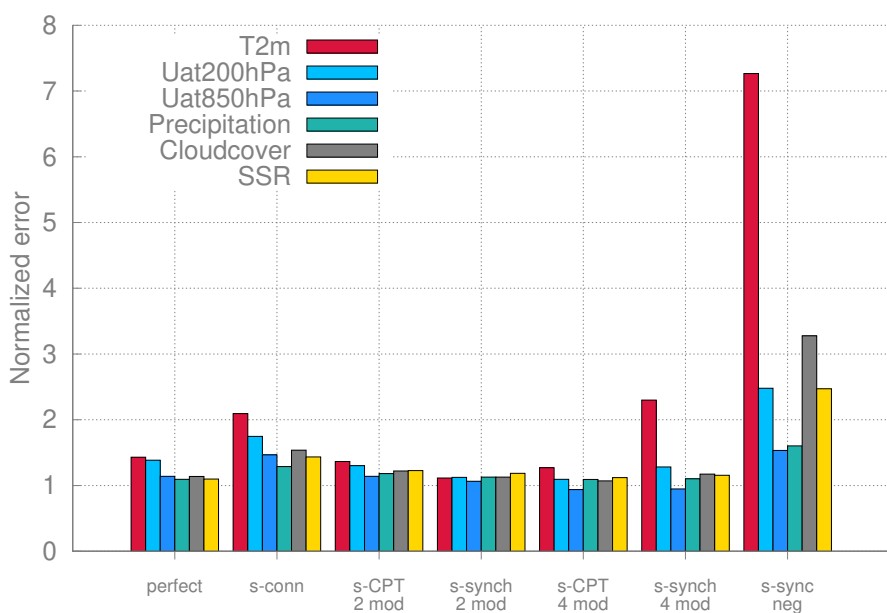

**Figure 15.** Overview the RMSE of the different supermodels (the connected supermodel of Selten et al. (2017) and the weighted supermodels from the experiments of this paper) over the model years 2011-2040 with respect to the truth.





10     of the synch rule is that it allows for negative weights that can potentially improve the weighted supermodel in case model errors do not compensate for positive weights. In addition, CPT requires fairly good models such that mixed trajectories are able to track an observed trajectory for some time. During the synch rule training on the other hand the nudging terms keep the supermodel in the neighborhood of the observed trajectory and is therefore more robust (i.e. less sensitive) with respect to the quality of the imperfect models.

15     In the application of CPT in this study, we encountered numerical issues due to the partial state replacement. A possible solution is the use of data assimilation techniques to combine state information from different models in a dynamical consistent manner (Asch et al., 2016; Carrassi et al., 2018). One straightforward solution along this line could be based on the idea of Du and Smith (2017), in which pseudo-orbit data assimilation is used instead of replacement of the entire state. Du and Smith (2017) have already used this approach successfully for low-order dynamical systems. These data assimilation techniques would also allow application of CPT in case the different models differ in state representation, for instance different numerical

grids.

    The weighted supermodels of this study have smaller climatological errors as compared to the connected supermodel based on the same two imperfect models in Selten et al. (2017). Also, in the four model experiment, the CPT supermodel has substantially better climatology than the supermodel trained by the synch rule. This suggests that synchronization with the truth can be difficult to obtain, especially when the imperfect models that form the supermodel are not fully synchronized

in case of a connected supermodel or when the weighted supermodel consists of several imperfect models. Although it is a common result in synchronization theory that identical systems will synchronize if the nudging strength is strong enough and if there are enough observations from the truth, in practice this can be a challenge. The issues with synchronized based learning can be easily demonstrated using a low-order dimensional system (not shown).

    In the second supermodel experiment of this paper the parameter perturbations of four imperfect models were chosen such

that they formed a so-called convex hull around the true parameter values. This implies that a linear combination with positive weights of these four models is able to reproduce the model equations with the true parameter values, provided that the parameters appear only linear in the equations. This is not exactly true in this case, but the trained weighted supermodel based on these four models turned out to have a climatology close to the truth. As all four imperfect models have a warmer and wetter climatology than the truth, simply taking the MME mean with positive weights thus does not improve the climatology. This

experiment is a clear example of the potential benefit of the supermodeling approach to ameliorate common model errors. This benefit arises due to the fact that model errors are compensated at an early stage, in the time-derivative, and not a posteriori, as in the MME approach where model errors have propagated spatially across the globe, across scales and across the different meteorological fields and other components of the climate system.

    In the final supermodel experiment we have explored the use of negative weights in order to improve predictions in the

case that model errors do not compensate, i.e. both imperfect models have parameter perturbations and climatological errors of the same sign. A supermodel trained using the synch rule yielded negative weights. With these weights, stable and credible simulations turn out to be possible and forecast errors as well as climatological errors are reduced with respect to the imperfect





models. Substantial errors remain as not all prognostic equations are combined (only temperature, vorticity and divergence, not humidity and surface pressure) and the parameters appear not linear in the equations.

Although the synch rule training does not impose that the weights sum to one, the training inevitably yielded sum of weights equal to one. An example based on the Lorenz 1963 equations (Lorenz, 1963) serves to illustrate why this might be the case. The Lorenz 1963 equations are:

$$
\begin{aligned}
\dot{x} &= \sigma(y-x) \\
\dot{y} &= x(\rho-z)-y \\
\dot{z} &= xy-\beta z,
\end{aligned}
\tag{8}
$$

where the standard parameter values are $\sigma = 10, \rho = 28$ and $\beta = \frac{8}{3}$. Assume we have two imperfect models with imperfect parameters $\rho_1$ and $\rho_2$. Then $\dot{y}_s = w_1(x(\rho_1 - z) - y) + w_2(x(\rho_2 - z) - y)$, with $s$ denoting the supermodel solution. We can rewrite this as: $\dot{y}_s = x(w_1\rho_1 + w_2\rho_2) - (w_1 + w_2)(xz + y)$. To reproduce the standard parameter model solution, two conditions must be satisfied: $(w_1\rho_1 + w_2\rho_2) = \rho$ and $w_1 + w_2 = 1$. Not only should the linear combination of imperfect parameter values match the true parameter value, also the weights have to sum to one.

The ultimate goal of our research is to apply supermodeling to realistic climate models. But, will it work? Based on the current results, we believe that this is possible, although the application is not as straightforward as for SPEEDO. First, state-of-the-art models are far bigger and more complex, making their numerical computation a substantial burden. This makes numerical efficiency a key aspect to consider. Second, the real world is not simply a perturbed parameter version of these complex models. In this paper we have worked under the hypothesis that model error only originates by error in the model

parameters. It is essential to extend the approach to other sources of model error towards the application with real climate models. In that case, on top of parametric error, model error can arise from the presence of unresolved scale, numerical discretization or incorrect physics.

Together with the realisms of the models (and of the related model error), that of the observations is also of central importance. In all previous studies with supermodeling, including the current, observations were assumed to be perfect, i.e. to

be complete and noisy free. To use real data it will thus be necessary to study the robustness of the supermodeling approach to noisy and unevenly distributed observations and to extend the methods to account for the observational noise. This latter problem is the subject of ongoing research of scientists which are making use of ideas and techniques from data assimilation. Data assimilation based supermodeling is also envisionable to account for generic source of model error in the construction of the supermodel, and it will be the subject of future research.

*Author contributions.* Francine J. Schevenhoven conceived the study, carried out the research and led the writing of the manuscript. Frank M. Selten provided codes and technical advice and provided with Alberto Carrassi and Noel Keenlyside input for the interpretation of the results and the writing.



*Competing interests.* The authors declare that they have no conflict of interest.

*Acknowledgements.* This work is partially supported by the EU project 648982 - STERCP (Synchronization To Enhance Reliability of Climate Predictions).



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
