# Peer review of "Improving weather and climate predictions by training of supermodels"

_Earth System Dynamics, 2019_

## Referee Comment (RC1) · Anonymous Referee #1 · 10 Jul 2019

This is a fine manuscript which needs only a few minor revisions before publication. The authors test two different methods of producing a weighted supermodel using versions of the SPEEDY atmospheric model with differing parametric settings in a coupled system (SPEEDO). The 'truth' model to which the supermodel is trained is also a version of SPEEDO with different parameter setting. The two different training methods used are Cross Pollination in Time and a variant of synchronization both of which produce weighted supermodels with more skill than the models which are combined. The manuscript details in a comprehensive manner the experiments run in both forecast and climate simulation mode along with the strengths and weaknesses of the experimental design. The discussion of weaknesses leads naturally to a discussion of next step experiments and open questions regarding the ultimate success of supermodeling

when trained to nature.

Suggested Minor Edits pg 2 line 1 replace 'hands' with 'hand' pg 2 line 25 replace 'evidences' with 'demonstrates' pg 3 line 29 replace 'complex' with 'complexity' pg 3 line 31 replace 'original' with 'new' or 'novel' pg 7 line 2 replace 'noisy' with 'noise' pg 22 line 4 replace 'become' with 'becoming' pg 22 line 5 replace 'precipitate' with 'precipitating' and delete 'achieve to' pg 25 line 4 replace 'linear' with 'linearly pg 25 line 13 replace 'Largest' with 'The largest' pg 25 Should ' sampling error' be replaced by 'natural variability' ? pg 28 line 29 replace 'appear not linear' with 'do not appear linearly' pg 29 line 20 replace 'noisy' with 'noise'
* * *

---

## Referee Comment (RC2) · Anonymous Referee #2 · 12 Sep 2019

General:

In the present paper the authors demonstrate the construction and training of a so called supermodel by means of a climate model of reduced complexity. Two training methods are applied and compared: Cross Pollination in Time (CPT), and synchronization based learning (synch rule). Both methods show their ability to construct a supermodel, which outperforms the individual (imperfect) models.

The combination of models to a supermodel appear to have significant potential to improve the skill of model predictions and projections. As discussed by the authors, supermodels may also be superior to commonly used multi model ensembles. Thus, a detailed description of the general idea, the methodology, and open questions is a very useful contribution. In my view, with this paper the authors succeed to give such

a demonstration. I very much like this paper, as it very clearly written and structured, and present more than enough new and valuable results to deserve publication. I have only two minor points mostly out of curiosity.

Minor:

1) Eq. 6c, synch role: it seems that the derivative of the model (f) with respect to a certain parameter (q) is needed for sync role. I may have overlooked it, but how is this derivative obtained. It somehow looked like one may have to use an adjoint (or at least linear) version of the model.

2) Coupling to the same ocean/land: I'm wondering how much the use of the same (perfect) ocean/land component for all models affect the training/behavior of the super-model, as, in my view, it may constrain the variability on long time scales to be similar in all models. In addition, the coupling to land/ocean may act similar (though more complex) to the nudging term (K(as-a0)). Perhaps, the authors have tried to build a supermodel from a set of totally independent SPEEDOs, and can comment on this.

---

## Author Response (AR1)

Authors response "Improving weather and climate predictions by training of supermodels" by Schevenhoven et al.

**Reply to Reviewer 1:**
We would like to thank Reviewer 1 for his/her positive comments. We revised our manuscript accordingly. We decided to use 'sampling error' instead of 'natural variability' since the differences between the averages of the runs provide an indication of the sampling error as a result of natural variability.

*Suggested Minor Edits pg 2 line 1 replace 'hands' with 'hand' pg 2 line 25 replace 'evidences' with 'demonstrates' pg 3 line 29 replace 'complex' with 'complexity' pg 3 line 31 replace 'original' with 'new' or 'novel' pg 7 line 2 replace 'noisy' with 'noise' pg 22 line 4 replace 'become' with 'becoming' pg 22 line 5 replace 'precipitate' with 'precipitating' and delete 'achieve to' pg 25 line 4 replace 'linear' with 'linearly pg 25 line 13 replace 'Largest' with 'The largest' pg 25 Should ' sampling error' be replaced by 'natural variability' ? pg 28 line 29 replace 'appear not linear' with 'do not appear linearly' pg 29 line 20 replace 'noisy' with 'noise'*

**Reply to Reviewer 2:**
We would like to thank Reviewer 2 for his/her helpful comments. We revised our manuscript accordingly and provide point-wise replies to the Reviewer below.

1. *Eq. 6c, synch role: it seems that the derivative of the model (f) with respect to a certain parameter (q) is needed for sync role. I may have overlooked it, but how is this derivative obtained. It somehow looked like one may have to use an adjoint (or at least linear) version of the model.*

In the paper the synch rule is applied to optimize the weights of the weighted supermodel. In our case $\mathbf{f}$ in Eq. 6c represents the supermodel tendency and $\mathbf{q}$ the weights. The derivative of $\mathbf{f}$ with respect to a certain weight is simply the tendency of the imperfect model belonging to that weight which is readily available. This because the supermodel tendency is a weighted superposition of the time derivatives of the imperfect models, as shown in Eq. 2c for an example of two imperfect models: $\dot{x}_s = W_1 \dot{x}_1 + W_2 \dot{x}_2$, where $\dot{x}_s$ denotes the supermodel tendency, $W_{1,2}$ the weights and $\dot{x}_{1,2}$ the imperfect model tendencies. Hence taking the derivative of the supermodel tendency with respect to the weight of an imperfect model results in the tendency of that imperfect model. We clarified this in the paper in section 4.2.

2. *Coupling to the same ocean/land: I'm wondering how much the use of the same (perfect) ocean/land component for all models affect the training/behavior of the super- model, as, in my view, it may constrain the variability on long time scales to be similar in all models. In addition, the coupling to land/ocean may act similar (though more complex) to the nudging term (K(as- a0)). Perhaps, the authors have tried to build a supermodel from a set of totally independent SPEEDOs, and can comment on this.*

Indeed, coupling to the same ocean/land constrains the variability near surface on longer time-scales in the different atmosphere models since the coupling acts as a nudging to the true ocean/land temperature through the surface heat fluxes during training. The reviewer asks how this affects the training of the supermodel and the behavior after training. In CPT training it helps to keep the CPT trajectory in the neighbourhood of the truth. A longer CPT trajectory allows for a more robust estimate of the weights. In case the imperfect models include imperfect representations of the land

and ocean components as well, a stronger drift from the truth can be expected during training since model errors in the evolution of the ocean and land state are introduced in addition to the model errors in the atmospheric evolution. We have not built a supermodel from a set of SPEEDOs with imperfections in the ocean and land components included. It remains to be seen whether in this more general case CPT training results in a weighted supermodel with improved prediction quality compared to the constituent imperfect models. Application of the synch rule in this more general case includes nudging of the ocean and land state to the truth so during training the supermodel cannot drift from the truth on longer timescales. But it also remains to be seen how much the long-term evolution of the supermodel can be improved on the basis of the minimization of the short-term synchronization errors in the presence of imperfections in the slow components of the climate systems. In this paper we have demonstrated that in the absence of imperfections in the slow components of the climate system, the long-term behavior of the supermodel improves while training only short-term prediction errors through the CPT or synch rule approach. A natural next step is to include imperfections in the slow components of the climate system as well.

**List of relevant changes in the revised manuscript**

- Page 11, line 16-19, point 1 Reviewer 2: A more clear explanation about the application of the synch rule for a weighted supermodel.
- Page 29, line 19-22, point 2 Reviewer 2: An explanation about a natural next step to include imperfections also 
[revised manuscript text omitted]